# Geographical Distribution and Genetic Diversity of Bank Vole Hepaciviruses in Europe

**DOI:** 10.3390/v13071258

**Published:** 2021-06-28

**Authors:** Julia Schneider, Bernd Hoffmann, Cristina Fevola, Marie Luisa Schmidt, Christian Imholt, Stefan Fischer, Frauke Ecke, Birger Hörnfeldt, Magnus Magnusson, Gert E. Olsson, Annapaola Rizzoli, Valentina Tagliapietra, Mario Chiari, Chantal Reusken, Elena Bužan, Maria Kazimirova, Michal Stanko, Thomas A. White, Daniela Reil, Anna Obiegala, Anna Meredith, Jan Felix Drexler, Sandra Essbauer, Heikki Henttonen, Jens Jacob, Heidi C. Hauffe, Martin Beer, Gerald Heckel, Rainer G. Ulrich

**Affiliations:** 1Institute of Novel and Emerging Infectious Diseases, Friedrich-Loeffler-Institut, Federal Research Institute for Animal Health, 17493 Greifswald-Insel Riems, Germany; marie.schmidt@charite.de (M.L.S.); stefan.fischer25@web.de (S.F.); 2Institute of Virology, Charité-Universitätsmedizin Berlin, 10117 Berlin, Germany; felix.drexler@charite.de; 3Institute of Diagnostic Virology, Friedrich-Loeffler-Institut, Federal Research Institute for Animal Health, 17493 Greifswald-Insel Riems, Germany; bernd.hoffmann@fli.de (B.H.); martin.beer@fli.de (M.B.); 4Research and Innovation Centre, Department of Biodiversity and Molecular Ecology, Fondazione Edmund Mach, 38098 San Michele all’Adige, Italy; fevolacri@gmail.com (C.F.); annapaola.rizzoli@fmach.it (A.R.); valentina.tagliapietra@fmach.it (V.T.); heidi.hauffe@fmach.it (H.C.H.); 5Department of Virology, Faculty of Medicine, University of Helsinki, 00100 Helsinki, Finland; 6Vertebrate Research, Federal Research Centre for Cultivated Plants, Institute for Plant Protection in Horticulture and Forests, Julius Kühn-Institute (JKI), Toppheideweg 88, 48161 Münster, Germany; christian.imholt@julius-kuehn.de (C.I.); Daniela.reil@uni-potsdam.de (D.R.); jens.jacob@julius-kuehn.de (J.J.); 7Department of Wildlife, Fish, and Environmental Studies, Swedish University of Agricultural Sciences, 90183 Umeå, Sweden; frauke.ecke@slu.se (F.E.); birger.hornfeldt@slu.se (B.H.); magnus.magnusson@skogsstyrelsen.se (M.M.); gert.e.olsson@lansstyrelsen.se (G.E.O.); 8Unit for Nature Conservation, County Administrative Board of Halland County, 30004 Halmstad, Sweden; 9Direzione Generale Welfare, U.O. Veterinaria, Piazza Città di Lombardia 1, 20124 Milan, Italy; Mario_Chiari@regione.lombardia.it; 10Centre for Infectious Disease Control, National Institute for Public Health and the Environment (RIVM), 3720 Bilthoven, The Netherlands; chantal.reusken@rivm.nl; 11Faculty of Mathematics, Natural Sciences and Information Technologies, University of Primorska, 6000 Koper, Slovenia; elena.buzan@famnit.upr.si; 12Environmental Protection College, 3320 Velenje, Slovenia; 13Institute of Zoology, Slovak Academy of Sciences (SAS), 81438 Bratislava, Slovakia; maria.kazimirova@savba.sk; 14Institute of Parasitology, Slovak Academy of Sciences, Hlinkova 3, 04001 Košice, Slovakia; stankom@saske.sk; 15Lancaster Environment Centre, Lancaster University, Lancaster LA2 0QZ, UK; tawhite201@gmail.com; 16Institute of Animal Hygiene and Veterinary Public Health, University of Leipzig, 04109 Leipzig, Germany; anna.obiegala@vetmed.uni-leipzig.de; 17Royal (Dick) School of Veterinary Studies and Roslin Institute, University of Edinburgh, Edinburgh EH8 9AB, UK; anna.meredith@unimelb.edu.au; 18Melbourne Veterinary School, Faculty of Veterinary and Agricultural Sciences, University of Melbourne, Melbourne, VIC 3052, Australia; 19Martsinovsky Institute of Medical Parasitology, Tropical and Vector-Borne Diseases, Sechenov University, 119991 Moscow, Russia; 20German Centre for Infection Research (DZIF), Associated Partner Site Berlin, 10117 Berlin, Germany; 21Department Virology and Rickettsiology, Bundeswehr Institute of Microbiology, 80937 Munich, Germany; sandraessbauer@bundeswehr.org; 22Natural Resources Institute Finland (LUKE), 00791 Helsinki, Finland; ext.heikki.henttonen@luke.fi; 23Institute of Ecology and Evolution, University of Bern, 3012 Bern, Switzerland; gerald.heckel@iee.unibe.ch

**Keywords:** bank vole hepaciviruses, HCV, *Hepacivirus F*, *Hepacivirus J*, rodent-borne pathogen, Europe, emerging virus

## Abstract

The development of new diagnostic methods resulted in the discovery of novel hepaciviruses in wild populations of the bank vole (*Myodes glareolus*, syn. *Clethrionomys glareolus*). The naturally infected voles demonstrate signs of hepatitis similar to those induced by hepatitis C virus (HCV) in humans. The aim of the present research was to investigate the geographical distribution of bank vole-associated hepaciviruses (BvHVs) and their genetic diversity in Europe. Real-time reverse transcription polymerase chain reaction (RT-qPCR) screening revealed BvHV RNA in 442 out of 1838 (24.0%) bank voles from nine European countries and in one of seven northern red-backed voles (*Myodes rutilus*, syn. *Clethrionomys rutilus*). BvHV RNA was not found in any other small mammal species (n = 23) tested here. Phylogenetic and isolation-by-distance analyses confirmed the occurrence of both BvHV species (*Hepacivirus F* and *Hepacivirus J*) and their sympatric occurrence at several trapping sites in two countries. The broad geographical distribution of BvHVs across Europe was associated with their presence in bank voles of different evolutionary lineages. The extensive geographical distribution and high levels of genetic diversity of BvHVs, as well as the high population fluctuations of bank voles and occasional commensalism in some parts of Europe warrant future studies on the zoonotic potential of BvHVs.

## 1. Introduction

The genus *Hepacivirus*, family *Flaviviridae*, comprises 14 species (International Committee on Taxonomy of Viruses, 2020, [1]) of enveloped viruses with an icosahedral nucleocapsid and a diameter of 50–65 nm. The single-stranded, positive-sense RNA genome with a length of approximately 9600 nucleotides (nt) encodes, in a single open reading frame, a polyprotein that is cleaved into ten proteins [2,3,4,5].

One of these hepaciviruses, the human hepatitis C virus (HCV), is one of the leading causes of liver cirrhosis and hepatocellular carcinoma [6]. The fact that this virus does not infect small mammals makes it challenging to find a suitable animal model [7,8]. However, the recent development of new diagnostic technologies has led to the identification of several new hepaciviruses in rodents, horses, cattle, dogs, and bats [9,10,11,12,13,14,15].

The bank vole is a rodent species inhabiting forests in large parts of Europe and southern parts of Western Siberia [16,17]. This species often shows multiannual fluctuations in population densities with peaks approximately every 3–4 years [18,19]. Phylogenetic analyses of bank voles (based on the mitochondrial cytochrome *b* gene, cyt *b*) revealed several evolutionary lineages, including Western, Eastern, Carpathian, Ural, and Italian, with different geographical distributions, likely caused by isolation of subpopulations in refugia during the last glacial period [20,21,22]. Interestingly, the current distribution of *Puumala orthohantavirus* (PUUV) largely follows this differentiation. In Central and Western Europe, PUUV is limited to the Western evolutionary lineage of the bank vole with few detections in Carpathian and Eastern lineage individuals in regions of sympatric occurrence with the Western lineage [23].

Molecular screening of rodents has resulted in the identification of two bank vole-associated hepacivirus (BvHV) clades [9]. The high genetic divergence of BvHV clade 1 and clade 2 led to their classification as distinct species within the genus *Hepacivirus*, namely, *Hepacivirus J* and *Hepacivirus F*, respectively. These viruses can be differentiated by real-time reverse transcription polymerase chain reaction (RT-qPCR) (Appendix A). The successful experimental infection of bank voles with BvHV has been proposed as the basis for the development of a small animal model for evaluation of human HCV infection [24].The objective of this study was to investigate the occurrence, geographical distribution, and genetic diversity of BvHV in bank voles and several other small mammal species from different geographic regions in Europe.

## 2. Materials and Methods

### 2.1. Collection of Small Mammals and Species Identification

Rodents and other small mammals were sampled in 14 European countries within the EU FP7 project EDENext and several national research projects [23,25,26,27,28,29] (Table 1; Figure 1). Rodent necropsy and tissue samplings followed previously established standard protocols. Morphological species determination for selected animals was confirmed by PCR and partial sequencing of the mitochondrial cyt *b* gene [30]. For evolutionary lineage identification of BvHV-infected bank voles, we selected 23 viral RNA-positive animals. Consensus sequences of a 764 nt fragment of *cyt b* gene were generated and included in a phylogenetic tree with reference sequences from different lineages of the bank vole [31]. The phylogenetic tree was constructed with MrBayes v3.2.7 [32], using a Markov chain Monte Carlo algorithm and the GTR+I+G substitution model, as determined using jModelTest v2.1.10 [33], and run for 4 × 10^6^ generations. Results were visualized using FigTree v1.4.4 (http://tree.bio.ed.ac.uk/software/figtree/) (Appendix A).

### 2.2. Nucleic Acid Extraction and RT-PCR Analyses

RNA extraction was performed using a phenol–chloroform protocol for liver or lung tissue samples of a large part of the sample collection, as previously described [25]. Another part of liver samples was homogenized in 500 µL phosphate-buffered saline (PBS). Here, RNA was extracted using Microlab Star automate (Hamilton Robotics, Reno, NV, USA) in combination with the Nucleo Spin 96 Virus Core Kit (Macherey-Nagel, Düren, Germany) according to the manufacturer’s instructions. For molecular screening, two different RT-qPCR assays were used for individual samples or for pools of six individuals: rodHCVeur assay [9] for *Hepacivirus J*, and RHV-NS3-Line4 assay [24] for *Hepacivirus F*. A subset of RT-qPCR-positive samples were then analyzed by conventional RT-PCR, targeting the NS3 gene, and amplification products were sequenced by dideoxy chain termination method (for primers, see Appendix A).

### 2.3. BvHV Sequence, Phylogenetic, and Isolation-by-Distance Analyses

A 472 nt fragment of the NS3 gene of the BvHV strains was aligned using Geneious Prime 2019.1.1 and MAFFT v7.388. A phylogenetic tree was constructed from the alignment using MrBayes v3.2.7 [35], with a Markov chain Monte Carlo algorithm, using the GTR+I+G substitution model, as determined using jModelTest v2.1.10 [33], run for 6 × 10^6^ generations; results were visualized with FigTree v1.4.4 (http://tree.bio.ed.ac.uk/software/figtree/).

A test for isolation-by-distance was performed including all BvHV sequences described here. Isolation-by-distance is the result of the accumulation of mutations in viral strains in local populations and spatially limited dispersal, and manifests as a statistical association between genetic differences and geographic distance [36]. Genetic distances between pairs of partial NS3 gene sequences from the study sites were estimated as p-distance using MEGA X [37]. For the isolation-by-distance analysis, geographic distances between the trapping locations were measured with the package geosphere v1.5-10 in the R (version 4) software environment [38].

### 2.4. Statistical Analysis

Mantel tests, as implemented in the ade4 (v1.7-16) package [39] in R (version 4) [38], were used to determine the statistical significance of the association between geographic and genetic distances.

A subset of the data was used to estimate the potential drivers of BvHV circulation within its host populations. Spatially and temporally replicated data were available for four regions in four federal states in Germany: Jeeser (Mecklenburg-Western Pomerania 54°9.75′ N, 13°15.55′ E), Gotha (Thuringia, 50°57.38′ N, 10°39.13′ E), Billerbeck (North Rhine-Westphalia, 51°59.63′ N, 7°18.99′ E), and Weissach (Baden-Wuerttemberg, 48°49.88′ N, 8°57.71′ E). In a generalized linear mixed effect model (with binomial error distribution), the individual infection status was used as a dependent variable, while individual factors (mass (g), sex (male/female), reproductive status (yes/no)) and population level (abundance of bank voles and co-occurring yellow-necked field mice (*Apodemus flavicollis*) as individuals per 100 trap nights, trapping season (spring/summer/autumn), and trapping year (2010/2011)) were treated as independent variables. The year 2012 was excluded from the analysis, as only spring data were available. Trapping location was incorporated as a random factor to account for the spatial design of the study. A multimodal inference approach was used to determine the most parsimonious model. First, the dredge function (MuMIn package, version 1.43.17) was used to rank all combinations of independent variables according to their conditional AIC (AICc). Second, from all combinations within a ΔAIC of <2 of the best model, the respective coefficients were averaged using the model.avg function. All analyses were performed in R (version 4) [38].

## 3. Results

### 3.1. Detection of BvHV RNA in Small Mammals from European Countries

The initial RT-qPCR screening of small mammals resulted in the detection of BvHV RNA in 442 out of 1838 (24.0%) bank voles from nine European countries and in one of seven (14.3%) northern red-backed voles from Finland, but in no other small mammal species (Table 1, Table 2 and Appendix A, and Figure 1). The highest site prevalence was detected in Vouzon (France) where 14 of 19 (73.7%) animals were positive in RT-qPCR (Table 2). Detailed temporal sampling of bank voles in the years 2010–2012 in four regions of Germany demonstrated the continuous circulation of BvHV in local populations (Table 3).

In total, 327 bank voles were positive in the rodHCVeur assay (*Hepacivirus J*), 202 bank voles and one northern red-backed vole were positive in the RHV-NS3-Line4 assay (*Hepacivirus F*), and 87 were positive in both assays (Table 2; for semiquantitative genome loads, see Appendix A). For 96 bank vole samples, partial NS3-gene sequences could be amplified and sequenced by RT-PCR; for 50 of these 96 vole samples consensus sequences were generated (available under GenBank accession numbers MW822235–MW822284). We observed a high background in electropherograms, and RT-PCR amplification failed on some RT-qPCR-positive samples with a low threshold cycle (Ct) value (e.g., Mu08/1265 Ct-value 18.75).

### 3.2. Sequence Divergence of BvHV

The use of two different RT-qPCR assays, partial sequencing of the NS3 gene, and phylogenetic analysis resulted in the confirmed detection of both BvHV species in bank voles. *Hepacivirus J* RNA was detected in voles from six countries, and *Hepacivirus F* RNA in animals from nine (Table 2, Figure 1, and Appendix A). Both BvHV species were detected by RT-qPCR in 87 bank voles from six countries (Table 2 and Figure 1); the occurrence of both BvHV species was confirmed by sequence and phylogenetic analysis in Germany and Slovakia (Figure 2).

A phylogenetic analysis confirmed the previously described [9] separation of clade 1/species *Hepacivirus J* and clade 2/species *Hepacivirus F* (Figure 2). The pairwise sequence variability at the nt level reached 21.2% and 24.3% within *Hepacivirus*
*J* and *Hepacivirus F*, respectively, and 48.2% between species. Amino acid sequence variability reached 8.6% and 12.6% within species, respectively, and 44.4% between species. Within trapping sites, nucleotide sequence divergences between BvHV strains ranged between 0 and 16.8% within species and 42.5% and 44.9 % between species (Appendix A). Both BvHV species independently showed very similar patterns of isolation-by-distance across their distribution ranges in Europe (Figure 3; Mantel tests; both *p* < 0.001). Pairwise genetic distances between sequences increased steeply within each BvHV species over relatively short geographic distances between trapping sites, and the increase levelled off for comparisons over larger geographic distances (Figure 3).

### 3.3. BvHV Association with Bank Vole Evolutionary Lineages

Phylogenetic analysis of cyt *b* sequences of infected bank voles indicated that at least four evolutionary lineages are susceptible to BvHV. Both virus species were detected in bank voles of the Western and Carpathian evolutionary lineages, and *Hepacivirus F* was also found in the Eastern lineage (Figure 2). BvHV RNA was also detected in a bank vole of the Ural lineage (trapping site: Umeå, Sweden). However, we were not able to generate a consensus BvHV sequence from this sample.

### 3.4. BvHV–Host Dynamics

Candidate models generated to evaluate BvHV circulation in bank voles as well as their respective AICc and model weights are presented in Table 4. A total of 14 factor combinations were within ΔAIC of <2 and considered for further model averaging. Relative importance (RI) of each factor can be approximated by the relative proportion of model weights including the respective factor (ranging from 0 (no importance) to 1 (very important)). The most important factor was “season” being selected in all candidate models (RI = 1), followed by “abundance of yellow-necked field mouse” (RI = 0.69), “mass” (RI = 0.46), “sex” (RI = 0.34), “reproductive activity” (RI = 0.18), “year” (RI = 0.10), and “abundance of bank voles” (RI = 0.05).

Averaged parameter estimates and their 95% confidence intervals are shown in Figure 4, where a factor can be considered significant when confidence intervals do not cross zero. Results indicate that “sex” or “reproductive activity” did not influence the infection status significantly. Although older individuals (as inferred from the “mass” factor) tended to have a higher infection risk (only positive estimates, Table 4), the effect size appeared to be small. The most dominant factor was “season”, with individuals caught in spring more likely to be infected. There was no significant effect of “bank vole abundance” on the infection risk, ruling out direct density-dependent transmission of BvHV. However, increasing abundance of yellow-necked field mouse tended to decrease individual infection risk in bank voles (only negative estimates, Table 4) though averaged confidence intervals still included zero (Figure 4).

## 4. Discussion

Two BvHV clades corresponding to species *Hepacivirus J* and *Hepacivirus F*, first described in 2013, were previously detected only in Germany and the Netherlands and later in the Ukraine [9,40]. In this study, we show a very wide geographical distribution of these viruses in Europe based on the detection of BvHV RNA in 442 bank voles from nine countries and one northern red-backed vole from Finland (Figure 1). The prevalence of BvHV in bank voles in many regions and the lack of detected viral RNA in the 23 other small mammal species suggest that the bank vole is one of the main reservoir hosts of these viruses. However, the sample size of many species was low, and future studies have to further investigate the possible susceptibility of these rodent species as potential hosts of the virus. The reservoir role of northern red-backed voles should be examined with additional samples from a broader region. These future studies might profit from adapted RT-qPCR protocols; the mismatches between the primer and probe sequences and the homologous BvHV species prototype strain might have caused a slightly reduced sensitivity of the RT-qPCR in our study (Appendix A).

We detected, for both BvHV species, a positive association between genetic differences and spatial distance. These isolation-by-distance patterns are similar to the patterns observed in bank vole-associated PUUV [36] or common vole-associated *Tula orthohantavirus* [41]. These associations can be explained by host dispersal occurring only at a local scale and the restriction of the accumulation of mutations in viral strains within local vole populations [42,43]. Over larger geographic distances, virus populations in each BvHV species thus evolve independently. However, in contrast to European hantaviruses [41,44,45], we detected both BvHV species in the same host populations in several regions of Europe, which suggests long-term co-existence of *Hepacivirus J* and *Hepacivirus F* in their host species.

The high genetic divergence between both hepacivirus species and their sympatric occurrence at multiple sites confirm their taxonomic classification as two distinct species. The nucleotide/amino sequence divergence within the NS3 gene/protein reached between both BvHV species a maximum level of 48.2%/12.6%. As this initial analysis described here is based on a short fragment of the NS3 gene, further studies are needed to examine the relationship of both BvHV species in the entire NS3 coding sequence/protein, several coding sequences/proteins, or the entire genome/polyprotein. In this study, we found that bank voles of four investigated evolutionary lineages, namely, Western, Carpathian, Eastern, and Ural, are susceptible to BvHV infection. Interestingly, both viruses were detected in the Western and Carpathian lineages of the bank vole, whereas *Hepacivirus F* can also infect the Eastern lineage (Figure 2). However, these initial findings, particularly for the Eastern and Ural lineages, are based on a small number of animals, and further studies need to be conducted to assess the relevance of each lineage.

The evaluation of individual and population-based factors did not show evidence for the influence of age, sex, or bank vole abundance on BvHV prevalence. This is in contrast to results obtained for PUUV and *Leptospira* prevalence in the same populations [26,45], where individual demographics can significantly determine individual infection risk. Our study did however find indications that increasing densities for non-hosts such as yellow-necked field mouse can potentially dilute BvHV in bank voles, similar to PUUV, for example, in Belgium [46].

High prevalences in spring have been reported for PUUV in Central Europe [47] and Scandinavia [48] and can be attributed to the contact of animals in their burrows during winter. This process entails that mainly older individuals in spring are infected and potentially new cohorts are still protected by transient immunity through maternal antibodies. The lack of age as a key factor in BvHV circulation might suggest a diverging transmission route of BvHV compared to PUUV. To our knowledge, there is no information on intraspecific transmission of hepaciviruses within wild rodent populations. In well-studied equine hepaciviruses, cases of vertical transmission during parturition have been described [49]. Theoretically, vertical transmission through blood transfer during pregnancy and/or parturition could explain why individual factors were not detected, which are often found associated in demographic analyses of more indirect, horizontally transmitted pathogens, like PUUV. This finding may also suggest an additional similarity of BvHV to human HCV, which is transmitted by blood, blood products and sexual contact or perinatally [50].

## 5. Conclusions

This study shows a broad geographical distribution of BvHV in bank voles across Europe. This is also reflected in the detection of BvHV RNA in bank voles of different evolutionary lineages. The high genetic divergence between both hepacivirus species and their sympatric occurrence in host populations at multiple sites clearly support their taxonomic classification as two distinct species, *Hepacivirus J* and *Hepacivirus F*. The influence of season on BvHV prevalence, but not of age, may suggest a varying transmission route (by vertical or direct contact) compared to other well-studied pathogens circulating in bank voles. Future investigations need to evaluate the role of additional lineages in the rodent host, for example, the Italian or Spanish ones, and the role of other *Myodes* vole species as possible reservoirs of BvHV, e.g., *M. rufocanus* and *M. centralis* in Asia or *M. gapperi* and *M. californicus* in North America. Finally, the multiannual fluctuation of bank vole populations warrants future investigations regarding the potential spillover of BvHV to other mammals and humans.

## Figures and Tables

**Figure 1 viruses-13-01258-f001:**
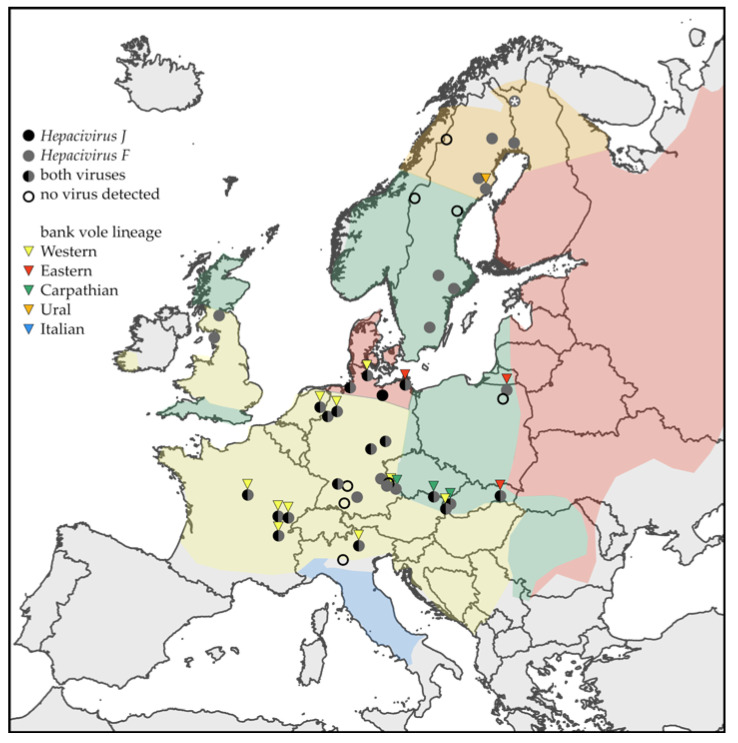
Map of bank vole and northern red-backed vole (*) trapping sites and the detection of *Hepacivirus J* (black) and *Hepacivirus F* (grey) in this study. Empty circles represent the lack of RNA detection of both hepacivirus species. Only trapping sites where five or more animals were sampled are shown. Colored areas correspond to the approximate distribution of the evolutionary lineages of the bank vole according to Filipi et al. [34], and colored triangles indicate evidence of the corresponding lineage in a bank vole in this study.

**Figure 2 viruses-13-01258-f002:**
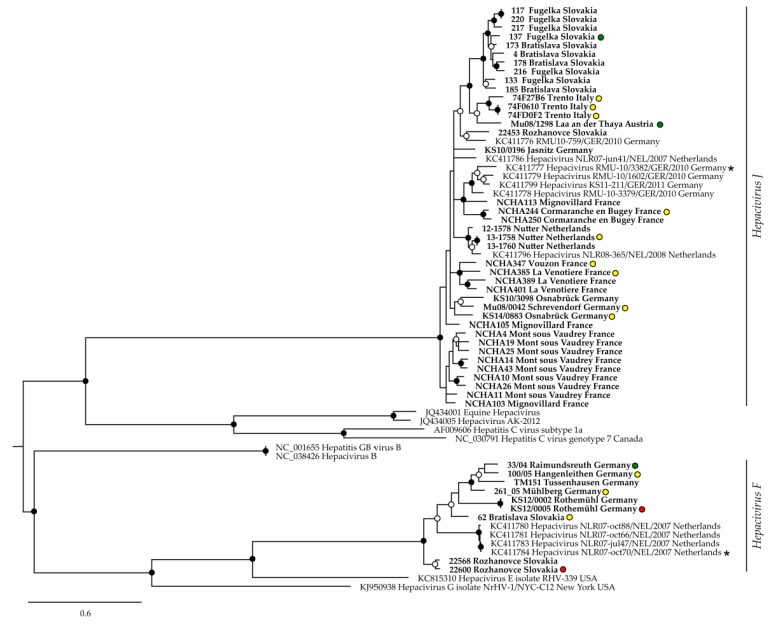
Phylogenetic relationships of partial BvHV NS3 sequences (472 nt). Sequences generated during this study are highlighted in bold. Circles at nodes indicate Bayesian posterior probabilities: black: probability exceeds 90%; white: probability exceeds 70%. The names of sequences obtained from GenBank include their accession numbers. Colored circles to the right of sequence names indicate the evolutionary lineage of the bank vole: yellow: Western lineage; red: Eastern lineage; green: Carpathian lineage. Asterisks mark type strains of both virus species.

**Figure 3 viruses-13-01258-f003:**
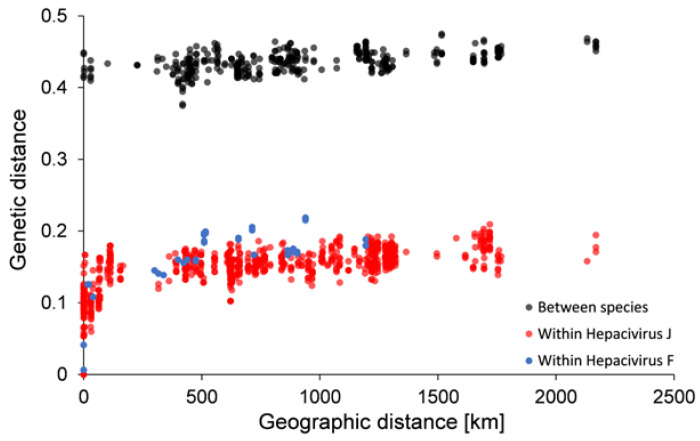
Relationships between pairwise genetic distance (*p*) and geographic distance between BvHV NS3 nucleotide sequences from bank voles collected across Europe. The virus species showed independently significant isolation-by-distance relationships (Mantel tests; both *p* < 0.001).

**Figure 4 viruses-13-01258-f004:**
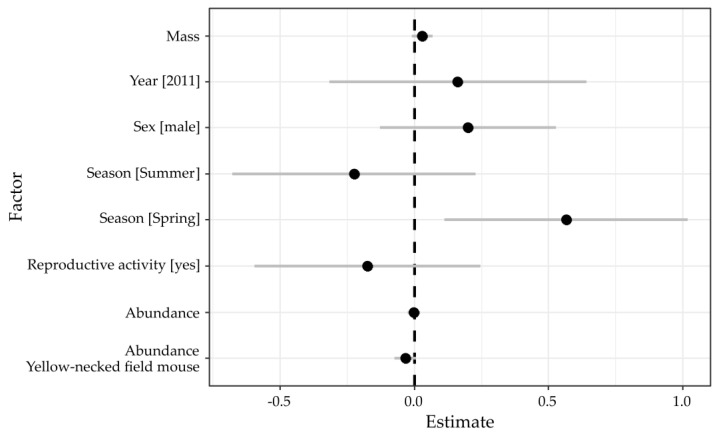
Graphical representation of the model averaging from all candidate models within an AIC of 2 of the best model (Models 1 to 14 in Table 4). Averaged mean estimates for all factors (black circle) and their 95% confidence interval (+/− grey line) are presented on the x-axis. For categorical factors, the reference categories are female (Sex), autumn (Season), the year 2010 (Year), and reproductive inactivity (No). Here, a factor can be considered significant if the confidence intervals do not include zero (dashed line).

**Table 1 viruses-13-01258-t001:** Overview of mammal species screened for presence of BvHV RNA and RT-qPCR results.

Order	Family	Species	No. of Hepacivirus-Positive/Tested Animals	% RT-qPCR Positive	Country (Total Number of Animals)
Rodentia	Cricetidae	*Lemmus lemmus*	0/21	0	FIN (21)
	Cricetidae	*Microtus agrestis*	0/8	0	GER (8)
	Cricetidae	*Microtus arvalis*	0/129	0	GER (129)
	Cricetidae	*Microtus oeconomus* (syn. *Alexandromys oeconomus*)	0/2	0	FIN (2)
	Cricetidae	*Myodes glareolus* (syn. *Clethrionomys glareolus*)	442/1838	24.0	GBR (61), GER (1297), FRA (99), ITA (25), NED (20), AUT (33), POL (15), SWE (223), SVK (65)
	Cricetidae	*Myodes rutilus* (syn. *Clethrionomys rutilus*)	1/7	14.3	FIN (7)
	Cricetidae	*Myopus schisticolor*	0/9	0	FIN (9)
	Muridae	*Apodemus* spp.	0/31	0	CRO (30), ITA (1)
	Muridae	*Apodemus agrarius*	0/5	0	CRO (2), GER (3)
	Muridae	*Apodemus flavicollis*	0/209	0	GER (206), ITA (3)
	Muridae	*Apodemus sylvaticus*	0/39	0	GER (39)
	Muridae	*Micromys minutus*	0/1	0	GER (1)
	Muridae	*Mus musculus*	0/22	0	GER (1), EST (8), FIN (3), LAT (8), LTU (2)
	Muridae	*Mus* spp.	0/2	0	CRO (2)
	Muridae	*Rattus* spp.	0/4	0	CRO (4)
Carnivora	Mustelidae	*Mustela nivalis*	0/1	0	ITA (1)
Eulipotyphla	Erinaceidae	*Erinaceus europaeus*	0/1	0	GER (1)
	Soricidae	*Crocidura leucodon*	0/1	0	ITA (1)
	Soricidae	*Crocidura russula*	0/1	0	GER (1)
	Soricidae	*Neomys fodiens*	0/1	0	ITA (1)
	Soricidae	*Sorex araneus*	0/37	0	GER (37)
	Soricidae	*Sorex alpinus*	0/1	0	ITA (1)
	Soricidae	*Sorex antinorii*	0/7	0	ITA (7)
	Soricidae	*Sorex coronatus*	0/26	0	GER (26)
	Soricidae	*Sorex minutus*	0/25	0	GER (25)
Total			443/2428	18.2%	

AUT, Austria; CRO, Croatia; EST, Estonia; FIN, Finland; FRA, France; GBR, Great Britain; GER, Germany; ITA, Italy; LAT, Latvia; LTU, Lithuania; NED, The Netherlands; POL, Poland; SVK, Slovakia; SWE, Sweden.

**Table 2 viruses-13-01258-t002:** Results of RT-qPCR assays of bank voles and northern red-backed voles (*) at country and trapping site level. All samples were tested in both assays.

Country	Trapping Site	rodHCVeur Assays (*Hepacivirus J*)	RHV-NS3-Line4 Assay (*Hepacivirus F*)	Both Assays (*Hepacivirus F* and *Hepacivirus J*)	Positive/Tested Per Trapping Site	Positive/Tested Per Country
Austria	Laa an der Thaya	3	10	1	12/33	12/33
Finland	Pallasjärvi	0 *	1 *	0 *	1/7 *	1/7 *
France	Cormaranche-en-Bugey	12	5	3	14/20	61/99
	La Venotiere	10	7	4	13/20	
	Mignovillard	9	5	3	11/20	
	Mont-sous-Vaudrey	9	2	2	9/20	
	Vouzon	8	12	6	14/19	
Germany	Ahlhorn	1	0	0	1/2	286/1297
	Bad Waldsee	0	0	0	0/5	
	Bierhütte	0	0	0	0/2	
	Billerbeck	67	8	0	75/285	
	Bogen	0	0	0	0/1	
	Bremerhaven	3	0	0	3/19	
	Falkenstein	0	2	0	2/5	
	Freyung	0	0	0	0/5	
	Geversdorf	0	0	0	0/4	
	Glashütte	0	0	0	0/1	
	Gotha	56	61	51	66/319	
	Hangenleithen	0	3	0	3/6	
	Jasnitz	5	0	0	5/20	
	Jeeser	41	3	2	42/155	
	Langenfurth	0	0	0	0/1	
	Lucka (bei Groitzsch)	3	1	0	4/17	
	Mühlberg, Spiegelau	0	1	0	1/1	
	Mutzenwinkel	0	1	0	1/2	
	Oberndorf (Hemmoor)	0	2	0	2/8	
	Osnabrück	11	8	4	15/23	
	Raimundsreuth	0	3	0	3/14	
	Reinberg	0	0	0	0/1	
	Rothemühl	0	2	0	2/4	
	Schrevendorf	9	2	0	11/20	
	Steinheim am Albuch	0	0	0	0/20	
	Treben (Altenburg)	0	0	0	0/3	
	Tussenhausen	0	4	0	4/9	
	Weissach	41	4	1	44/332	
	Wolbrechtshausen	1	0	0	1/2	
	Wolfertschlag	0	1	0	1/5	
	Zußdorf	0	0	0	0/5	
	Zwiesel	0	0	0	0/1	
Great Britain	Cumbria	0	2	0	2/51	5/61
	Pentland Hills	0	3	0	5/10	
Italy	Brescia	0	0	0	0/5	5/25
	Trento	3	2	0	5/20	
The Netherlands	Nutter	11	4	2	13/20	13/20
Poland	Mikołajki	0	0	0	0/5	1/15
	Dobskie island	0	1	0	1/5	
	Dejguny island	0	0	0	0/5	
Slovakia	Bratislava	8	9	1	16/22	42/65
	Fugelka	10	7	3	14/23	
	Rozhanovce	6	10	4	12/20	
Sweden	Ammarnäs	0	0	0	0/20	17/223
	Grimsö	0	4	0	4/20	
	Haparanda	0	2	0	2/20	
	Harads	0	3	0	3/20	
	Öster Malma	0	1	0	1/20	
	Umeå	0	2	0	2/42	
	Vålådalen	0	0	0	0/20	
	Västernorrland	0	0	0	0/20	
	Växjö	0	4	0	4/21	
	Vindeln	0	1	0	1/20	
Total		327	202 + 1 *	87		

**Table 3 viruses-13-01258-t003:** Detection of BvHV RNA in bank voles collected at four sites in Germany from 2010 to 2012.

Year	Season	Number of BvHV RNA-Positive/Total Number of Bank Voles							
		Jeeser		Billerbeck		Gotha		Weissach	
2010	spring	5/11	26/82	24/84	57/242	8/48	23/218	15/131	24/262
	summer	8/28		15/84		6/65		5/77	
	autumn	13/43		18/74		9/105		4/54	
2011	spring	0/1	9/59	1/1	7/32	3/6	22/89	0/0	5/31
	summer	0/23		3/17		10/45		5/23	
	autumn	9/35		3/14		9/38		0/8	
2012	spring	6/15		3/11		10/15		10/39	
Total		41/156		67/285		55/322		39/332	

**Table 4 viruses-13-01258-t004:** Binomial generalized linear models explaining the probability of BvHV infections in bank voles. Presence of a factor within a candidate model is indicated either by a + for categorical factors or the estimate for continuous factors. If a factor is not included in a candidate model, the cell remains empty. Models with ΔAIC > 2 were excluded from subsequent model averaging. AIC of the null model was 958.7 and was therefore not included in the Table below. df = degrees of freedom; logLik = log-likelihood value.

Model	Factors							Model Statistics				
	Population				Individual							
	Abundance (Bank Vole)	Abundance (Yellow-Necked Field Mouse)	Season	Year	Reproductive Activity	Sex	Mass	df	logLik	AICc	ΔAICc	Model Weight
1		−0.0339	+					5	−468.021	946.1	0	0.06
2		−0.0327	+				0.0248	6	−467.137	946.4	0.26	0.053
3		−0.0335	+			+	0.0270	7	−466.347	946.8	0.7	0.042
4		−0.0346	+			+		6	−467.379	946.8	0.74	0.041
5			+					4	−469.513	947.1	0.96	0.037
6			+				0.0263	5	−468.525	947.1	1.01	0.036
7		−0.0350	+		+		0.0349	7	−466.604	947.3	1.22	0.033
8		−0.0360	+		+	+	0.0378	8	−465.743	947.6	1.53	0.028
9			+			+	0.0284	6	−467.808	947.7	1.6	0.027
10			+	+				5	−468.927	947.9	1.81	0.024
11			+			+		5	−468.936	947.9	1.83	0.024
12		−0.0306	+	+				6	−467.967	948	1.92	0.023
13		−0.0346	+		+			6	−467.978	948	1.94	0.023
14	−0.0021	−0.0320	+					6	−468.004	948.1	1.99	0.022
15	−0.0101		+					5	−469.044	948.1	2.05	0.022

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
