# Peer review of "Geographical Distribution and Genetic Diversity of Bank Vole Hepaciviruses in Europe"

_viruses, 2021, doi:10.3390/v13071258_

Round 1

Reviewer 1 Report

The manuscript by Schneider et al. describes the molecular epidemiology of vole hepaciviruses in Europe. Viruses from 2 different species were investigated and samples from several countries and small mammal species were screened. I find the study very well done and the analyses are very thorough. The results are interesting, conclusions are supported by the results, and the manuscript is well written.

My only major concern is that no sampling approved protocol numbers, research sampling permits, or animal care approvals are provided (unless I missed it). I am assuming they are there as samples have been used for other publications, they just have to be specified.

Other comments:

- Do you have any evidence for disease in these animals. Were there abnormalities detected during necropsies?

- I think the biggest study limitation is that a very short fragment was used for genetic analyses and this should be clearly stated in the discussion.

- Line 291. You only have a few animals typed as Carpathian and Eastern lineages. I’d say it is worth mentioning this here as a limitation for this analysis.

- By the cartoon in Figure S1, the primers and probe used to detect species F viruses do have many mismatches compared to the reference sequence. Maybe this could cause your positivity rates to be underestimated. You should probably comment on that.

Minor:

- At line 99 you mention a “consensus sequence of 764 nt in length”, but the caption of Figure S3 says it was 464. Please check this.

- Lines 111-12. There is something wrong with this sentence.

- The primers reported in Table S1 seem to have been designed to amplify the whole NS3, but only a fragment of ~500nt was sequenced. Were full NS3 sequences acquired but not used for the analyses?

- Lines 132-4. Was this p-distance or you used a genetic distance model for your estimates? This has to be specified.

- Table 3. What does that “NA” refer to?

- Figure 2. You might want to provide a higher definition version because text is poorly readable.

- Lines 284-7. You state: “Problems observed in electropherograms, failure to generate consensus sequences, or failure in RT-PCR amplification of RT-qPCR-positive samples with a low Ct value, may indicate additional sequence variability or the occurrence of quasispecies, similar to that known for HCV [46].” However, this is only partially true. Failure in amplification could be the result of primer mismatch, while background noise could be due to non-specific amplification. I suggest removing this sentence because it unnecessarily opens a can of worms and some degree of sequencing failure is expected and totally normal for these kinds of studies.

- Lines 316-7. I am not sure what you mean. This sentence is not really clear, I suggest rephrasing.

Author Response

Reply to referees comments │viruses-1244550

Reviewer #1

1.) The manuscript by Schneider et al. describes the molecular epidemiology of vole hepaciviruses in Europe. Viruses from 2 different species were investigated and samples from several countries and small mammal species were screened. I find the study very well done and the analyses are very thorough. The results are interesting, conclusions are supported by the results, and the manuscript is well written.

Reply:  

We would like to thank referee #1 for thorough evaluation of the manuscript. Below is a point-by-point reply to the referee’s comments and a description of modifications to the manuscript according to the suggestions.

My only major concern is that no sampling approved protocol numbers, research sampling permits, or animal care approvals are provided (unless I missed it). I am assuming they are there as samples have been used for other publications, they just have to be specified.

Reply:  

As mentioned in the manuscript, rodents and other small mammals were sampled in 14 European countries within the EU FP7 project EDENext and several national research projects. The details on the sampling permits and approvals have been already given in a previous manuscript (Fevola et al., 2020, Vector Borne Zoonotic Dis., reference #25), but, as suggested by the reviewer, we added at the end of the manuscript (“Ethical permits”):

“Small mammal trapping was performed with permission from the ethical committees in the respective countries according to their national laws [25]: Czech Republic: authorized in the protocol PP 27/2007 (institutional and state committees of the Czech Academy of Sciences in 2007); France: authorized under French and European regulations on care and protection of laboratory animals: French Law 2001-486 issued on June 6, 2001 and Directive 2010/63/EU issued on September 22, 2010; all animal procedures (trapping, euthanasia) were pre-approved by the Direction des Services Vétérinaires of the Herault Department under Agreement B 34-169-1; Finland: snap trapping does not require ethical permits under the Finnish Act on Animal Experimentation 62/2006 and by the decision of Finnish Animal Experiment Board 16 May 2007. A permit (23/5713/2001) for capturing protected species (Sorex spp., Myodes rufocanus and Myopus schisticolor) was granted by

the Finnish Ministry of the Environment; Germany: Small mammals were trapped using Sherman©live animal traps according to relevant legislation (H. B. Sherman Traps Inc., Tallahassee, Florida, U.S.A.) (official permit Site R1: Regierung der Oberpfalz 55.1-8646.4-140, Site T: Regierung von Schwaben 55.1-8646-2/30, Site S: AZ 36.11-36.45.12/4/12-001). Additional sample collection was authorized according to relevant legislation and by permission of the federal authorities (permits Regierungspräsidium Stuttgart 35-9185.82/0261, Landesamt für Natur, Umwelt und Verbraucherschutz Nordrhein-Westfalen 8.87-51.05.20.09.210, Landesamt für Landwirtschaft, Lebensmittelsicherheit und Fischerei Mecklenburg-Vorpommern 7221.3-030/09, Thüringer Landesamt für Lebensmittelsicherheit und Verbraucherschutz 22-2684-04-15-107/09; Italy: authorized by the ‘Comitato Faunistico Provinciale della Provincia di Trento’, protocol n° 595, issued on 04 May 2011; The Netherlands: authorized in compliance with Dutch laws on animal handling and welfare: RIVM/DEC permits 200700119, 200800053, 200800113 and 20100139; Poland: approved by the First Local Bioethical Committee in Krakow (decision # 48/2007); Slovakia: authorized according to current laws of the Slovak Republic, approved by the Ministry of Environment of the Slovak Republic, licence numbers 297/108/06-3.1, 6743/2008-2.1 and ZPO-594/2012-SAB; Slovenia: authorized by the Ministry of Culture of the Republic of Croatia (No. 532–08–01-01/1–11-03) and the Veterinary Administration of the Republic of Slovenia (No. 34401–36/2012/9; Sweden: authorized under the Animal Ethics Committees of Umeå: A 44-08, A 61-11, and A 121-11, the Swedish Board of Agriculture: A 135-12 and Dnr A78-08, and the Swedish Environmental Protection Agency: Dnr 412-2635-05, Dnr 412-4009-10, Nv 02939-11). A Home Office Project Licence (PPL 60/3652) under the Animals (Scientific Procedures) Act 1986 were obtained, to authorise restraint, anaesthesia and blood sampling of live wild rodents in Great Britain. The sampling was also approved by the local (Royal (Dick) School of Veterinary Studies) and institutional (University of Edinburgh) Ethical Review Process. Additional bank voles were collected in Germany and Austria by pet cats, during pest rodent control or by forest institutions due to their official duties that did not require a special permit.

(end of reply)

2.) Other comments:

Do you have any evidence for disease in these animals. Were there abnormalities detected during necropsies?

Reply:  

Unfortunately, the animal dissections were not made by pathologists; therefore, we do not have any information about abnormalities.

(end of reply)

3.) I think the biggest study limitation is that a very short fragment was used for genetic analyses and this should be clearly stated in the discussion.

Reply:

We followed this suggestion and mentioned this limitation within the discussion chapter, which reads as follows:

“The nucleotide/amino sequence divergence within the NS3 gene/protein reached between both BvHV species a maximum level of 48.2 %/12.6 %. As this initial analysis described here is based on a short fragment of the NS3 gene, further studies are needed to examine the relationship of both BvHV species on the entire NS3 gene/protein, several genes/proteins or the entire genome/polyprotein.”

(end of reply)

4.) Line 291. You only have a few animals typed as Carpathian and Eastern lineages. I’d say it is worth mentioning this here as a limitation for this analysis.

Reply:

We added information in the discussion, which reads now as follows:

“In this study, we found that bank voles of four investigated evolutionary lineages, namely Western, Carpathian, Eastern, and Ural, are susceptible to BvHV infection. Interestingly, RNA of both viruses was detected in the Western and Carpathian lineage of the bank vole, whereas Hepacivirus F can also infect the Eastern lineage (Figure 2). However, these initial findings, particularly for the Eastern and Ural lineages, are based on a small number of animals and further studies need to be conducted to assess the relevance of each lineage.”

(end of reply)

5.) By the cartoon in Figure S1, the primers and probe used to detect species F viruses do have many mismatches compared to the reference sequence. Maybe this could cause your positivity rates to be underestimated. You should probably comment on that.

Reply:

Indeed, there are several mismatches between the Hepacivirus F prototype strain and the selected primer and probe sequences. The selection of these primer and probe sequences was based in a sequence comparison of different BvHV strains. Some of these differences are caused by degenerated primers; to make that more clear we added an explanation for residues M, A or C; R, A or G; S, G or C; W, A or T; Y, T or C. In addition, the 5´ and 3´ ends of all 4 primers and both probes fit to 100% to the homologous prototype virus sequence. Nevertheless, we could not exclude a potential influence of the mismatches to the result of our study and added the following text to the discussion:

“The mismatches between the primer and probe sequences and the homologous BvHV species prototype strain might have caused a slightly reduced sensitivity of the real-time RT-PCR.”

(end of reply)

6.) Minor:
At line 99 you mention a “consensus sequence of 764 nt in length”, but the caption of Figure S3 says it was 464. Please check this.

Reply:    

We corrected the caption of Figure S3. As mentioned in the main text the analysed fragment of the cyt b gene had a length of 764 nt.

(end of reply)

7.) Lines 111-12. There is something wrong with this sentence.

Reply:   

We corrected this sentence that now reads as follows:

“RNA extraction was performed using a phenol-chloroform protocol for liver or lung tissue samples of a large part of the sample collection, as previously described [25].”

(end of reply)

8.) The primers reported in Table S1 seem to have been designed to amplify the whole NS3, but only a fragment of ~500nt was sequenced. Were full NS3 sequences acquired but not used for the analyses?

Reply:    

Indeed, we aimed to amplify the whole NS3 gene, which was not successful for all samples likely due to the high virus diversity. Therefore, we decided to restrict this study to a partial NS3 gene segment, which worked for most animals in order to be able to analyze animals from a broader geographical area. We removed the redundant primer sequences from Table S1.

(end of reply)

9.) Lines 132-4. Was this p-distance or you used a genetic distance model for your estimates? This has to be specified.

Reply:    

We used p-distances for our analyses. This is now specified in the Methods section 2.3. and reads as follows:

“Genetic distances between pairs of partial NS3 gene sequences from the study sites were estimated as p-distances using MEGA X [36].”

(end of reply)

10.) Table 3. What does that “NA” refer to?

Reply:    

“NA” was from a previous version of the Table. We removed it.

(end of reply)

11.) Figure 2. You might want to provide a higher definition version because text is poorly readable.

Reply:    

We provided Figure 2 in a higher resolution.

(end of reply)

(end of reply)

12.) Lines 284-7. You state: “Problems observed in electropherograms, failure to generate consensus sequences, or failure in RT-PCR amplification of RT-qPCR-positive samples with a low Ct value, may indicate additional sequence variability or the occurrence of quasispecies, similar to that known for HCV [46].” However, this is only partially true. Failure in amplification could be the result of primer mismatch, while background noise could be due to non-specific amplification. I suggest removing this sentence because it unnecessarily opens a can of worms and some degree of sequencing failure is expected and totally normal for these kinds of studies.

Reply:    

We agree and removed the sentence.

(end of reply)

13.) Lines 316-7. I am not sure what you mean. This sentence is not really clear, I suggest rephrasing.

Reply:    

We modified the sentence that reads now as follows:

“The high genetic divergence between both hepacivirus species and their sympatric occurrence at multiple sites confirms their taxonomic classification as two distinct species."

(end of reply)

Reviewer 2 Report

The manuscript by Julia Schneider and co-authors shows with an extremely rich set of data the geographical distribution of bank vole-associated hepaciviruses (BvHV) and their genetic diversity in Europe. The presentation of the results, the evaluation and the discussion are to a high degree consistent. The paper is structured logically and reads fluently.

Even if the results shown here do not come from a classical hypothesis-based approach, they do lay an important basis for the investigation of the variance and specificity of hosts, viral infectious agents and secondary factors.

Here are just a few small specific comments:

  • It's not entirely clear to me why Figure S2 is needed in addition to Figure 1.For me there is no scientific added value here.
  • Even if there is no consensus in the English language on whether or not to insert a space between the number and the percent sign, it should at least be uniform. At least the International System of Units declares "a space separates the number and the symbol %". That's why I prefer this variant.

Author Response

Reply to referees comments │viruses-1244550

The manuscript by Julia Schneider and co-authors shows with an extremely rich set of data the geographical distribution of bank vole-associated hepaciviruses (BvHV) and their genetic diversity in Europe. The presentation of the results, the evaluation and the discussion are to a high degree consistent. The paper is structured logically and reads fluently.

Even if the results shown here do not come from a classical hypothesis-based approach, they do lay an important basis for the investigation of the variance and specificity of hosts, viral infectious agents and secondary factors.

Here are just a few small specific comments:

Reply:    

We’d like to thank referee #2 for thorough evaluation of the manuscript. Below is a point-by-point reply to the referee’s comments and a description of modifications to the manuscript according to the suggestions.

(end of reply)

1.) It's not entirely clear to me why Figure S2 is needed in addition to Figure 1.For me there is no scientific added value here.

Reply:    

In contrast to Figure 1 (which only shows bank vole and Northern red-backed vole trapping sites), Figure S2 shows all trapping sites of this study including those where no bank voles were captured. This includes and adds trapping sites in Croatia, Estland, Latvia, and Lithuania. However, we followed the suggestion of the reviewer and deleted Figure S2 as the information in this Figure is also provided by Table 1 and Table S2.

(end of reply)

2.) Even if there is no consensus in the English language on whether or not to insert a space between the number and the percent sign, it should at least be uniform. At least the International System of Units declares "a space separates the number and the symbol %". That's why I prefer this variant.

Reply:    

We agree and added the missing spaces.

(end of reply).

Reviewer 3 Report

The study by Schneider et. al. provides a deep dive into the distribution and genetic diversity of hepaciviruses in bank voles, across multiple sampling sites throughout Europe. The data presented shows a broad geographical distribution of BvHV, including Hepachivirus J and Hepachivirus F, in multiple host lineages. Using several analysis methods, they also indicate a link between genetic differences of BvHV and geographical distances. 

The study is well designed, performed and written. English language is good also. I have only a few very minor comments and recommend this paper for publication. 

Line 124: In which mode/algorithm was MAFFT used, please indicate. 

Line 135: Please indicate a version for geosphere. Does it need a reference? Perhaps with the "citation" function if needed. 

Line 135: Please indicate the base R version used. 

Line 138 and beyond: I think also here for all the packages a version number and reference would be good (i.e. for the ade4 and MuMIn packages) 

Line 169: I am not sure what the (*) refers to in the legend of this figure, maybe I am missing something. 

Table 3: Something seems off with the formatting of this table, maybe some table headings lost their bold-typeface during manuscript editing. 

Author Response

Reply to referees comments │viruses-1244550

The study by Schneider et. al. provides a deep dive into the distribution and genetic diversity of hepaciviruses in bank voles, across multiple sampling sites throughout Europe. The data presented shows a broad geographical distribution of BvHV, including Hepachivirus J and Hepachivirus F, in multiple host lineages. Using several analysis methods, they also indicate a link between genetic differences of BvHV and geographical distances.

The study is well designed, performed and written. English language is good also. I have only a few very minor comments and recommend this paper for publication.

Reply:

We’d like to thank referee #3 for thorough evaluation of the manuscript. Below is a point-by-point reply to the referee’s comments and a description of modifications to the manuscript according to the suggestions.

(end of reply)

1.) Line 124: In which mode/algorithm was MAFFT used, please indicate.

Reply:

For aligning, we used the MAFFT plugin of Geneious and chose the “Auto” algorithm with default setting. The algorithm selects an appropriate strategy from L-INS-i, FFT-NS-i and FFT-NS-2 according to data size.

(end of reply)

2.) Line 135: Please indicate a version for geosphere. Does it need a reference? Perhaps with the "citation" function if needed.

Reply:

We used geosphere 1.5-10. We added this information in the text, which now reads as follows:

“For the isolation-by-distance analysis, geographic distances between the trapping locations were measured with the package geosphere v1.5-10 in the R (version 4) software environment [37].”

 (end of reply)

3.) Line 135: Please indicate the base R version used.

Reply:

We used R version 4, we added this information in the text:.

“Second, from all combinations within a Δ AIC of <2 of the best model, the respective co-efficients were averaged using the model.avg function. All analyses were performed in R (version 4) [37]. “

 (end of reply)

4.) Line 138 and beyond: I think also here for all the packages a version number and reference would be good (i.e. for the ade4 and MuMIn packages)

Reply:    

We added information, which now reads as follows:

“Mantel tests, as implemented in the ade4 (v1.7-16) package [38] in R (version 4) [37], were used to determine the statistical significance of the association between geographic and genetic distances.”

“A multimodal inference approach was used to determine the most parsimonious model. First, the dredge function (MuMIn package, version 1.43.17) was used to rank all combi-nations of independent variables according to their conditional AIC (AICc).”

(end of reply)

5.) Line 169: I am not sure what the (*) refers to in the legend of this figure, maybe I am missing something.

Reply:    

In Figure 1, we show the trapping sites of bank voles. However, as we also detected viral RNA in one out of seven northern red-backed voles, we wanted to show the corresponding trapping site as well. To enable the distinction of both vole species trapping sites, we use the asterisk to mark this single trapping site of northern red-backed voles.

(end of reply)

6.) Table 3: Something seems off with the formatting of this table, maybe some table headings lost their bold-typeface during manuscript editing.

Reply:    

We appreciate the comment and corrected the formatting of Table 3.

(end of reply)

Reviewer 4 Report

The manuscript by Schneider et al. "Geographical distribution and genetic diversity of bank vole hepaciviruses in Europe" analyzes the distribution of 2 hepacivirus species (J and F) in small rodent population in Europe. The study is well designed and performed in its multicentric approach and succeeds in clarifying the epidemiolgy of these viruses in European myodes spp. populations. Sampling sizes for other small rodents, in particular for urban-dwelling small rodent populations are scant and this may be a limit for the overall scientific relevance of this study.

Author Response

Reply to referees comments │viruses-1244550

The manuscript by Schneider et al. "Geographical distribution and genetic diversity of bank vole hepaciviruses in Europe" analyzes the distribution of 2 hepacivirus species (J and F) in small rodent population in Europe. The study is well designed and performed in its multicentric approach and succeeds in clarifying the epidemiolgy of these viruses in European myodes spp. populations. Sampling sizes for other small rodents, in particular for urban-dwelling small rodent populations are scant and this may be a limit for the overall scientific relevance of this study.

Reply: 

We’d like to thank referee #4 for thorough evaluation of the manuscript. Below is a point-by-point reply to the referee’s comments and a description of modifications to the manuscript according to the suggestions.

We agree that the number of rodent species other than Myodes spp. is quite low. To address this issue, we modified the discussion, which reads now as follows:

“In this study, we show a very wide geographical distribution of these viruses in Europe based on the detection of BvHV RNA in 442 bank voles from nine countries and one northern red-backed vole from Finland (Figure 1). The prevalence of BvHV in bank voles in many regions and the lack of detected viral RNA in the 23 other small mammal species suggest that the bank vole is one of the main reservoir hosts of these viruses. However, the sample size of many species was low and future studies have to further investigate the possible susceptibility of these rodent species as potential hosts of the virus. The reservoir role of northern red-backed voles should be examined with additional samples from a broader region.”